# Alginate Based Core–Shell Capsules Production through Coextrusion Methods: Recent Applications

**DOI:** 10.3390/foods12091788

**Published:** 2023-04-25

**Authors:** Chanez Bennacef, Stéphane Desobry, Laurent Probst, Sylvie Desobry-Banon

**Affiliations:** 1Laboratoire d’Ingénierie des Biomolécules (LIBio), ENSAIA-Université de Lorraine, 2 Avenue de la Forêt de Haye, BP 20163, 54505 Vandoeuvre-lès-Nancy Cedex, France; chanez.bennacef@univ-lorraine.fr (C.B.); sylvie.desobry@univ-lorraine.fr (S.D.-B.); 2Cookal SAS Company, 19 Avenue de la Meurthe, 54320 Maxéville, France

**Keywords:** encapsulation, hydrogel, oil, probiotics, food

## Abstract

Encapsulation is used in various industries to protect active molecules and control the release of the encapsulated materials. One of the structures that can be obtained using coextrusion encapsulation methods is the core–shell capsule. This review focuses on coextrusion encapsulation applications for the preservation of oils and essential oils, probiotics, and other bioactives. This technology isolates actives from the external environment, enhances their stability, and allows their controlled release. Coextrusion offers a valuable means of preserving active molecules by reducing oxidation processes, limiting the evaporation of volatile compounds, isolating some nutrients or drugs with undesired taste, or stabilizing probiotics to increase their shelf life. Being environmentally friendly, coextrusion offers significant application opportunities for the pharmaceutical, food, and agriculture sectors.

## 1. Introduction

Alginate is a biopolymer extracted from brown algae and some bacteria and is composed of guluronate and mannuronate units (G and M units, respectively) [1]. The polymer is a versatile and safe material that is widely used in various industries such as cosmetic, textile, pharmaceutical, and food as a coating material, stabilizer, thickener, and disintegrating agent [2]. It is generally recognized as safe (GRAS), due to its non-toxic, non-antigenic, biocompatible, and biodegradable properties [3]. The ability of alginate to form a hydrogel through ionic cross-links with divalent and trivalent ions (generally, calcium ions) makes it a preferred choice for the encapsulation technology, especially for extrusion and coextrusion methods [4]. The composition of alginate varies among different algae species, and its gelation is influenced by factors such as molecular weight, percentage of M and G units, and its concentration, in addition to various experimental conditions. Therefore, high-molecular-weight and high-G-content alginate allows obtaining high-viscosity hydrogels and thus compact and rigid beads and capsules [5]. However, excessive viscosity can cause nozzle clogging and produce non-spherical capsules [4]. On the other side, a high percentage of M units will lead to an elastic alginate network [6]. To enhance the functional properties of alginate, researchers are exploring various modifications and composite gels made with the addition of other polysaccharides and proteins [7,8]. These modifications aim to improve the encapsulation efficiency, payload, storage stability, and barrier properties of alginate.

Encapsulation is a flexible and adaptable technology that is utilized in various industries, including the pharmaceutical, food, and agriculture industries [9,10,11]. One of the primary reasons for using encapsulation is to protect and control the release of the encapsulated material. This technology involves enclosing small particles or droplets within a protective coating or shell, which can prevent them from being degraded or affected by external factors such as temperature, humidity, and light. This protective coating can also enable the targeted delivery of the encapsulated material, allowing it to be released at a specific time or location [12]. Additionally, encapsulation can enhance the stability and shelf life of products, improve their sensory properties, and reduce their volatility or reactivity [13]. Various structures can be obtained by encapsulation, i.e., particles, powders, capsules, beads, and core–shell capsules.

Core–shell capsule production is possible utilizing a single nozzle for the encapsulation process. This type of encapsulation involves gelation reactions that result in cross-linking only occurring in the area around the interface between the “inside” and the “outside” of the capsule [14,15]. This technique is known as reverse spherification and is used in many industries, such as bubble tea production. However, these products are subject to content diffusion from the core through a very fragile and very porous shell [16].

On the other hand, the coextrusion encapsulation method allows the production of core–shell capsules by injecting a core solution and a shell solution through a concentric nozzle. The shell material and core material pass through outer and inner orifices, respectively [17]. The shell in this system presents a higher mechanical resistance [18] and lower core diffusion than reverse-spherification core–shell systems. Therefore, we chose to focus on coextrusion systems in this review.

Additionally, the core–shell system allows a higher payload [15,19], than other encapsulation technologies, such as spray–drying and simple extrusion that usually produce powders and solid particles. Such formulations are more likely to lose their content payload and stability due to external environment exposure [20].

Numerous reviews treating encapsulation technologies and methods have been published recently focusing on food applications [21,22] or cell encapsulation and biomedical applications [23,24]. However, to our knowledge, no review has been published on coextrusion encapsulation and its recent applications.

## 2. Oil and Essential Oil Encapsulation

Besides the food and nutraceutical industries, oils are commonly used in cosmetics for fragrance or in plant-based extracts [25,26], agricultural pesticides or insecticides [27,28], and pharmaceutical active molecules [29,30]. Numerous papers describe oil and essential oil encapsulation using different coextrusion methods and process parameters. These have been adapted to each experimental environment to successfully produce capsules with a wide size range of 300–2000 µm and various payloads, as shown in Table 1. For these examples, oil is injected through the internal nozzle, and alginate, either with or without copolymers, through the external annular nozzle, as illustrated in Figure 1. 

### 2.1. Reduced Oxidation

In order to avoid lipid oxidation and deterioration, the most common strategy is the exclusion of initiators and promotors of lipid oxidation. Lipids are usually stored under vacuum or inert gas in opaque and non-metallic containers [48]. However, oil encapsulation is a great approach to isolate oil from the external environment. Besides the protective function and stability enhancement, oil encapsulation presents many others benefits such as taste, color, and odor masking, many studies described various oil encapsulation as illustrated in Table 1. An encapsulated oil can also allow the controlled release of liposoluble molecules [23]. For some applications such as in the food and cosmetic industries, the encapsulated oil presents a solid form (spray–dried oil). This physical conversion allows its incorporation in various products such as yoghurts, sausages, and various power-based products [49,50,51].

Core–shell oil capsules present a high oil content depending on the oil, shell material, and encapsulation parameters used in their production. As shown in Table 1, it appears that a higher core flow rate will result in the entrapment of a greater amount of core material and engender capsule production with high encapsulation efficiency (up to 99.2% with an oil load of 82.8%) [42].

Alginate as a shell material for oil encapsulation is probably one of the most used polymers in coextrusion technology. Its use as a unique shell material exhibited good results for various oils encapsulation, and some predictive mathematical models for oil capsules properties have been reported [18,35]. The addition of other polymers to alginate can improve the oil load as observed for low-/high-amylose-content starch where a high canola oil content has been reported (58%) [44]. The addition of soy protein isolates and HMP resulted also in high load values, i.e., 83 and 95%, respectively. Other additional shell materials have been explored, including hydroxypropyl methylcellulose (HPMC), agar, fish gelatin, and κ-carrageenan. They exhibited interesting results in combination with alginate for oil capsule production [39,52]. The addition of chitosan as an additional coating material for kenaf seed oil and ginger oil alginate capsules permitted to reduce the shell porosity [37,39]. This was similarly observed for soy protein isolates added to alginate [38].

Several research works have studied the oxidative stability of different oils, with or without antioxidant addition in the hydrophobic phase and with various shell material composition [32,41,47], Oil oxidation was generally evaluated through the determination of the peroxide value (PV) and the anisidine value (p-AV) that indicate primary oxidation and secondary oxidation, respectively [37,49]. According to Table 2, most oils encapsulated by coextrusion had reduced values of oxidation indicators, and for most of them, less bioactives’ loss was reported. Effective reduction of oxidation was observed for linseed and rapeseed oil encapsulated with chitosan or alginate, where alginate/linseed oil capsules and chitosan/linseed oil capsules presented PV < 48.6 meq/kg and PV < 7.49 meq/kg, respectively, after 4 weeks at 40 °C compared to PV > 65 meq/kg for free linseed oil. Secondary oxidation indicators were also lower for encapsulated-oil alginate/linseed oil capsules and chitosan/linseed oil capsules, which showed p-AV < 5.06 and p-AV < 3.03 respectively, while free linseed oil had p-AV > 9.33. Similar results were obtained for encapsulated rapeseed oil, where alginate/rapeseed oil capsules and chitosan/rapeseed oil capsules presented PV < 20.77 meq/kg and PV < 19.92 meq/kg, respectively, after 4 weeks at 40 °C compared to PV > 65 meq/kg for free rapeseed oil; indicators for secondary oxidation were also lower for encapsulated rapeseed oil, as shown in Table 2 [37]. 

Better results against oxidation were obtained with chitosan coating than with uncoated alginate/polymers capsules [46,47,53]. Reduced kenaf seed oil oxidation was achieved as well as low phytosterol and tocopherol losses after 24 days of storage at 65 °C for alginate–HMP–chitosan shells [41,42,46]. HMP enhanced shell and oil stability [46] and exhibited great results combined with alginate for roselle seed oil encapsulation [40]. 

To enhance the oil stability against oxidation, some authors studied the influence of antioxidant addition in the oily phase, i.e., caffeic acid, phloridzin, butylated hydroxytoluene (BHT), quercetin, and vitamin E. Caffeic acid addition (300 ppm) provided a better protection against the oxidation of olive oil, though not against hydrolytic rancidity. In addition, caffeic acid improved monounsaturated and polyunsaturated fatty acids protection [31]. 

Encapsulated canola oil fortified with quercetin presented great results for primary oxidation [44], similar to those obtained with BHT, which is no longer used due to its toxicity [54]. Capsules with quercetin addition in the core material presented a higher phenolic content after storage but a lower oil stability [32].

### 2.2. Reduced Evaporation and Release

Essential oils represent good candidates for encapsulation also as they contain highly volatile and oxygen-sensitive functional molecules [55]. The core–shell structure provided by coextrusion methods should then be efficient to protect them from alteration and evaporation [33]. Rosemary essential oil has been successfully encapsulated by coextrusion with alginate as the shell material and still presented great results in antimicrobial activity (bacteria and fungi reduction) after processing. Further, the prolonged release conferred by the encapsulated form increased the essential oil shelf life and bioactivity [33].

Some other beneficial properties are provided by encapsulation. For instance, in vitro digestion used to simulate human gastrointestinal digestion demonstrated great results in terms of protection of kenaf seed oil encapsulated in alginate and chitosan capsules [46]. A good absorption of the encapsulated biomolecules in the duodenum was observed after in vitro digestion [45]. Furthermore, soy protein isolate introduction into alginate increased the shrinkage degree of β-carotene capsules and delayed β-carotene release in the stomach [38].

### 2.3. Reduced Taste

Taste is important in determining the success of oral formulations’ commercialization. Several drugs and nutraceuticals have an unpleasant taste, especially in liquid dosage forms. To overcome this issue, they are often formulated with flavors and sweeteners to mask the bitterness associated with the active and inactive ingredients [56]. In this way, the bitter taste reduction can enhance the acceptability and adherence to medication and complements. Encapsulation is one of the most effective methods that are used for taste masking, especially of methods using an additional external coating to cover undesirable tastes and flavors [57]. Therefore, the core–shell structure is a great candidate for this purpose. Core–shell oil capsules allow the consumption of hydrophobic antioxidants and biomolecules with bitter taste at a high concentration, at which they present the greatest effectiveness and bioactivity [58]. The fortification of encapsulated oil in core–shell capsules facilitates the oral intake of such compounds [32].

## 3. Probiotics Encapsulation

Probiotics are described as viable microorganisms that can provide health benefits, largely described in the literature, such as maintaining the gut microbiota, improving digestion, reducing lactose intolerance, lowering serum cholesterol, inhibiting pathogen growth, and preventing certain cancers [59,60]. These effects are observed if microorganisms are alive and in a sufficient amount (10^6^–10^9^ CFU/g) [61,62]. Generally, probiotics are categorized as food supplements and can be divided in (i) probiotics for foods, including foods, food ingredients, and dietary supplements, (ii) probiotics for drugs that are used as a cure and for disease treatment or prevention, (iii) designed probiotics (genetically modified probiotics), and (iv) feed probiotics, which are used for animals [63,64].

Probiotics are very sensitive to damaging conditions, and their viability is highly impacted by environmental parameters, i.e., pH, temperature, water activity (a_w_), storage conditions, and processing [61,65]. Consequently, multiple encapsulation methods have been employed for the protection and controlled release of probiotics [66,67].

Spray-drying [68], emulsions [69], and electrohydrodynamic atomization [70] have been used to produce probiotics particles for food matrices. Nonetheless, a lot of studies reported a lower viability of spray-dried probiotics due to thermal, osmotic, and oxidative stress induced by the process conditions [71]. The extrusion technology does not require high temperatures and organic solvents; thus, it is of high interest for cells and probiotics encapsulation [71]. The coextrusion process producing reservoir-type particles can stabilize and isolate the probiotics from the surroundings during storage and delivery into the human body [59,65,72]. In addition to its protective role, probiotics encapsulation facilitates their controlled release across the intestinal tract [73]. Probiotic core–shell capsules are produced using the coaxial nozzle system with adaptation due to sterile environment requirements; the process is illustrated in Figure 2.

However, the encapsulation effectiveness in keeping probiotics alive depends on many factors, i.e., material type and concentration used for immobilization, encapsulation method, capsule size, and physicochemical characteristics of the environment [74].

Numerous materials have been evaluated for probiotics encapsulation. The most used is alginate for its biocompatibility and cheapness [75], but its use is limited due to its high porosity and sensitivity to acidic environments [5]. The addition of copolymers to the alginate shell, as the blend alginate–shellac, presented promising results in porosity reduction [61]. Table 3 shows alginate–polymers combinations that have been used recently. In addition, chitosan-coated alginate beads showed a denser structure with great protective properties due to their good resistance to the deteriorating and chelating effect of calcium [65]. 

Pectin was also successfully used in combination with alginate. Blends improved probiotics viability and preservation from the environment [74]. Additionally, pectin from fruit can also be incorporated as an effective prebiotic, enhancing probiotics growth and activity [76,77]. Overall, as shown in Table 3, probiotics survivability in simulated gastrointestinal conditions was significantly higher after their encapsulation. Combining prebiotics, i.e., pectin, mannitol, maltodextrin, inulin, and fructo-oligosaccharides (FOS), enhances probiotics survival in the upper gastrointestinal tract and improves their benefits due to a synergistic health effect [59,78,79]. 

Therefore, due to the increasing health awareness of consumers, the demand for functional foods containing probiotics is constantly growing [80,81]. So far, most probiotics can be found in dairy products, and their consumption is in constant increase. However, lactose-intolerant, allergic, vegan, and vegetarian populations are in continuous growth, which introduces the need to develop dairy-free probiotic products, i.e., milk-less beverages of plant origin and functional foods [59]. Therefore, encapsulation is a great way to incorporate probiotics into any food matrices. Beverages such as herbal teas and fruit juices enriched with probiotics presented promising results as functional food candidates [59,80,82]. Core–shell capsules of lactobacillus acidophilus NCFM have been successfully incorporated in mulberry tea. The shell was made of alginate and mannitol as a prebiotic and provided stability for 4 weeks, with a probiotic concentration higher than the minimum required (6 log10 CFU/mL) [59].

**Table 3 foods-12-01788-t003:** Encapsulation efficiency and survivability of probiotics encapsulated in various shell materials using coextrusion.

Shell and/or Coating Material	Probiotics	Prebiotics	Parameters	Capsules Size (µm)	EE (%)	SGC (ph, Duration)	Survivability (%)		Refs
							Encapsulated Probiotics	Free Probiotics	
Alginate	*Lactobacillus acidophilus*	Apple skin Polyphenols	Frequency: 2723 Hz	423–486	96.7	SGF (pH 2, 2 h)	73.9	54.1	[80]
Alginate	*Lactiplantibacillus plantarum*	Inulin	Inner/outer nozzle: 150/300 µm, Frequency: 300 Hz, Air pressure: 600 mbar	685	95	SGF (pH 2, 2 h) SIF (pH 7.5, 5 h)	92.0 97.5	88.4 94.5	[65]
Alginate	*Bifidobacterium lactis Bi-07*	Galacto-oligosaccharides	Inner/outer nozzle: 200/300 µm, Frequency: 300 Hz, Air pressure: 600 mbar	736	94	SGF (pH 2, 2 h) SIF (pH 7.5, 5 h)	91.3 82	80.5 75.0	[83]
Alginate–low-methoxylated	*Bifidobacterium infantis*	/	Inner/outer nozzle:150/300 µm	520–568	ND	ND	ND	ND	[74]
Alginate/pectin	*Lactobacillus rhamnosus GG*	Black bean extract	Inner/outer nozzle: 150/300 µm, Frequency: 500 mL, Air pressure: 300 Hz	715	98.3	SGF (pH 2, 2 h) SIF (pH 7.4, 4 h)	94.1 79.9	90.1 62.5	[72]
Alginate/chitosan	*Lactobacillus plantarum 299v*	Oligofructose	Inner/outer nozzle: 150/300 µm, Air pressure: 600 mbar, Frequency: 300 Hz	648–790	93.0	SGF (pH 2, 2 h) SIF (pH 7.5, 5 h)	97.0 85.0	80.0 75.0	[73]
Alginate–chitosan	*Bifidobacterium animalis subsp. lactis BB-12*	Mannitol	Inner/outer nozzle: 200/300 µm Air pressure: 600 mbar, Frequency: 300 Hz	800	89.2	SGF (pH 2, 2 h) SIF (pH 7.5, 3 h)	97.3 86.7	74 97.5	[84]
Alginate–chitosan	*Lactobacillus plantarum 299v*	/	Inner/outer nozzle: 150/300 µm Air pressure: 600 mbar, Frequency: 300 Hz	620	97.7	SGF (pH 2, 2 h) SIF (pH 7.2, 4 h)	97.0 95.5	93.5 0.0	[85]
Alginate– chitosan	*Lactobacillus rhamnosus GG*	Flaxseed mucilage	Inner/outer nozzle: 200/300 µm, Inner/outer rate: 1.0/7.8 mL/min	780	98.8	SGF (pH 2, 2 h) SIF (pH 7.0, 4 h)	84.9 93.2	35.4 47.9	[82]
Alginate– chitosan	*Lactobacillus acidophilus 5*	Isomalto-oligosaccharide	Inner/outer nozzle: 200/300 µm, Air pressure: 600 mbar, Frequency: 300 Hz	616	92.2	SGF (pH 2, 2 h) SIF (pH 7.2, 2 h)	60.9 0.0	0.0 0.0	[86]
Alginate– alginate/shellac	*Lactobacillus paracasei BGP-1 in sunflower oil/coconut fat*	/	Inner/outer nozzle: 450/700 µm, Inner/outer rate: 2/16 mL/min, Frequency: 100 Hz	710–860	ND	SGF (pH 1.8, 2 h) SIF (pH 6.5, 3 h)	95.0 100	50.0 70.0	[61]
Alginate–Alginate/Shellac	*Lactobacillus acidophilus LA3 in sunflower oil*	/	Inner/outer nozzle: 450/700 µm, Inner/outer rate: 2/16 mL/min Frequency: 100 Hz	690–760	ND	SGF (pH 1.8, 2 h) SIF (pH 6.5, 3 h)	92.0 93.0	ND ND	[71]
Alginate–Alginate/Poly-L-lysine	*Lactobacillus rhamnosus GG*	Isomalto-oligosaccharide	Inner/outer nozzle: 200/300 µm, Air pressure: 600 mbar, Frequency: 300 Hz	491–541	90.4	SGF (pH 2, 2 h) SIF (pH 7.5, 2 h)	75.0 0.0	39.2 0.0	[78]
Alginate/locust bean gum	*Lactobacillus acidophilus NCFM*	Mannitol	Inner/outer nozzle: 200/300 µm, Air pressure: 600 mbar, Frequency: 300 Hz	550–700	96.8	SGF (pH 2, 2 h) SIF (pH 7.5, 3 h)	78.4 70.2	64.9 0.0	[59]

ND: not determined, SGC: simulated gastric conditions, SGJ: simulated gastric juice, SIJ: simulated intestinal juice, EE: encapsulation efficiency, Refs: references.

## 4. Encapsulation of Other Ingredients

In addition to oil and probiotic encapsulation, extrusion has been used for the encapsulation of various other ingredients. For instance, cells encapsulation using the coextrusion technology is one of the most promising methods for three-dimensional cell culture that mimics in vivo the three-dimensional structure of tissues and organs [87].

The core–shell structure appears to be superior to other bead structures for long-term cell culture and for three-dimensional self-assembled cellular structures formation [88]. A semipermeable shell is necessary to provide a favorable microenvironment for the cells, with excellent permeability for mass transfer [87,88,89]. It ensures minimal solute diffusion and peripheral cells’ escape through the shell. Considering the self-organization capacities of human pluripotent stem cells, Cohen et al. (2023) successfully engineered a bioreactor based on a coextrusion milli-fluidic system [90]. The system provided 3D micro-compartments in core–shell capsules, with controlled size, seeding cell density, and oxygen, as illustrated in Figure 3. Additionally, polyethyleneimine incorporation in the hydrogel shell reduced capsule swelling, resulting in improved stability and minimal protein adsorption [89]. A similar system for producing 3D cellular assays to study tumor progression have been previously described [91]. The method involves encapsulating and growing cells inside permeable, elastic, hollow microspheres. These spheres can serve as mechanical sensors to measure the pressure exerted by the expanding spheroid. It is then possible to investigate the dynamics of pressure buildup and its effects on cellular density and organization within the spheroid [91]. Research on invasion assays in a collagen matrix has suggested that mechanical cues from the surrounding microenvironment may trigger cell invasion from a growing tumor [91]. Coextrusion offers a unique avenue to produce in vitro cell-based assays useful for developing new anticancer therapies and investigating the interplay between mechanics and growth in tumor evolution.

Furthermore, individual compartments provided by core–shell capsules are a great way to control a coculture behavior, e.g., in host–microbiome studies, through intestinal cells and bacteria encapsulation [87]. Promising results have been reported for cell encapsulation using the coextrusion technology. Stem cells and multipotent stromal cells have been effectively encapsulated by coextrusion. Great cell viability, metabolic activity, and cell–cell interactions were achieved. Cancer cells (CT26, Caco-2) have also been successfully encapsulated in core–shell capsules with a thin shell; in this case, the addition of sodium dodecyl sulfate surfactant was required for capsule production [92]. An adequate core viscosity ensures the flowability, and low-viscosity high-G alginate improved in vitro culture stability and promoted pluripotency in multipotent stromal cells. Furthermore, the M/G ratio in alginate appeared to impact the encapsulated cells [88]. Core–shell capsules showed a high potential as advanced functional carriers for stem cells and organoids’ efficient cultivation and transplantation and are a promising tool for tissue engineering and regenerative systems for medical applications [93,94].

The coextrusion technology is also promising for core–shell carrier system production for the simultaneous encapsulation of multiple active substances in a single carrier. The system is solvent-free and allows a precise control of particle size and composition. Highly efficient encapsulation and controlled release of two synergistic anticancer drugs were achieved. In this study, 0% of cancerous human cells viability was reached after 20 h of incubation with the drugs doxorubicin hydrochloride and paclitaxel [95].

Core–shell liquid microcapsules presented great results for pharmaceutical extraction from drinking water. Whelehan et al. (2010) used liquid-core microcapsules loaded with dibutyl sebacate and oleic acid and obtained promising results for the rapid extraction of diclofenac, metoprolol, warfarin, and carbamazepine from water [96].

Capsules are suitable, as reactive cells, for fatty acid methyl esters production, which is an alternative to diesel fuel. Modified alginate capsules were used as a micro-reactor for transesterifying triglycerides and esterifying free fatty acids [97].

The core–shell structure system is also a great candidate for phase-change materials encapsulation. Capsules of n-nonadecane have been successfully produced with the electro-coextrusion process for thermo-regulating textiles. These 200–400 µm microcapsules contained up to 84% of n-nonadecane and had an energy-storing density over 137 J·g^−1^ [98].

## 5. Conclusions

For many years, encapsulation has been utilized across multiple fields and for diverse applications. Encapsulation can be employed to shield the enclosed substance from its external environment, enhancing the stability of a dispersion, preventing denaturation and chemical reactions with external compounds, as well as slowing down the oxidation and evaporation of the encapsulated substance. Encapsulation also facilitates the controlled release of the enclosed substance, which can be released with the degradation of the encapsulating material at a desired time. For this purpose, actives are encapsulated in various forms, i.e., powder, liposomes, beads, core–shell capsules. This function is particularly useful as it allows utilizing these objects as carriers and functionalizing them with active substances in cosmetic or pharmaceutical applications.

Nonetheless, core–shell capsules offer several advantages over simple extrusion processes. Firstly, they allow for a large payload due to their reservoir-type structure, enabling a higher quantity of active ingredients to be encapsulated, which is a very important feature for encapsulation efficiency and oil encapsulation payload, as discussed above. Secondly, the shell provides a protective barrier for the encapsulated ingredient, improving its stability against oxidation and limiting the evaporation of highly volatile actives such as essential oils. Thirdly, the shell also enables slow-release kinetics, regulating the release of the encapsulated substance over time or enabling a targeted release to allow actives, e.g., probiotics, to be protected from degradation in the gastrointestinal tract. Additionally, core–shell capsules can produce a burst-like effect when broken, resulting in a high release of flavor, for example, in chewing gum. The continuous production process of core–shell capsules is also an added advantage.

Overall, the use of certain polymer/copolymers matrices for shell solutions enhances the shell properties, both for oil protection against oxidation and for probiotics protection during gastric digestion. The coextrusion technology is a very versatile method, in addition it is applicable for the encapsulation of a large variety of materials (oils, dyes, probiotics, cells, peptides, etc.). It can be conducted in a sterile environment and is scalable for industrial production. The encapsulation of complex ingredients or viable cells and at a large industrial scale is developing to answer the wide requests in the food, medical, pharmacological, and environmental fields.

## Figures and Tables

**Figure 1 foods-12-01788-f001:**
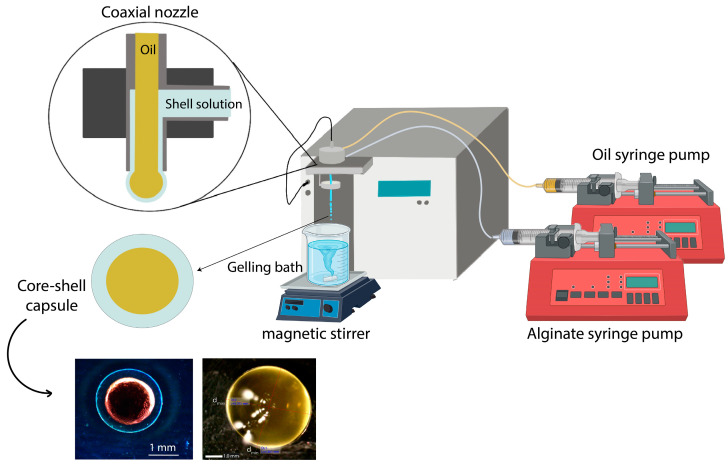
Illustration of oil encapsulation in core–shell capsules using the coextrusion method.

**Figure 2 foods-12-01788-f002:**
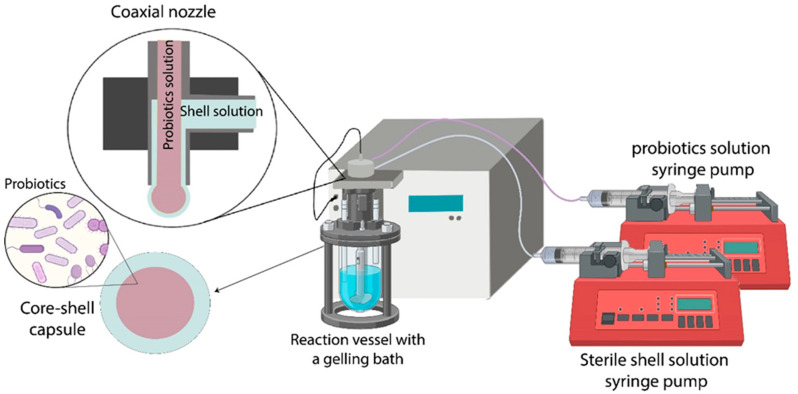
Illustration of probiotics encapsulation in core–shell capsules using the coextrusion method.

**Figure 3 foods-12-01788-f003:**
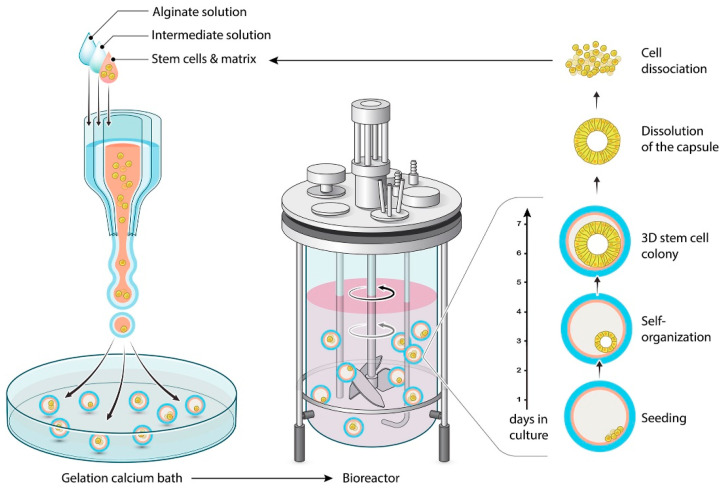
Encapsulation and scale-independent culture of encapsulated 3D human pluripotent stem cell colonies in bioreactors [90].

**Table 1 foods-12-01788-t001:** Characteristics of alginate core–shell capsules produced by coextrusion methods.

Shell and/or Coating Material	Oil/Essential Oil	Key Parameters of the Coextrusion Method	Average Capsule Size (µm)	Treatment Post Encapsulation	EE (%)	Load (%)	References
Alginate	Olive	Frequency: 1706 Hz	ND	None	ND	ND	[31]
Alginate	Canola	Inner/Outer rate: 30/200 mL/h Outer nozzle: 200 µm Frequency: 1750 Hz	400	pH treatment, freeze drying	ND	55.2	[32]
Alginate	Rosemary EO	Inner/Outer nozzle: 200/400 µm Inner/Outer rate: 900/300 mL/h Frequency: 350 Hz	950 (756 dried)	Oven-drying (60 °C for 2 h)	ND	ND	[33]
Alginate	Fish oil	Frequency: 1300 Hz	600	None	ND	ND	[34]
Alginate	Sunflower	Inner/Outer nozzle: 750/900 µm Inner/Outer rate: 14,55/34 mL/min.	2060	None	ND	37	[35]
Alginate	Sunflower (emulsion with CaCl_2)_	Inner/Outer nozzle: 400/600 µm	800	None	ND	ND	[36,37]
Alginate	Linseed	Dripping height: 5 cm	ND	None	91	38.4	[37]
Alginate	Rapeseed	Dripping height: 5 cm	ND	None	91.6	39.9	[37]
Alginate–Soy protein isolate	Sunflower + β-carotene	Inner/Outer nozzle: 150/300 µm Inner/Outer rate: 120/400 mL/h Frequency: 1000 Hz	567	None	99.2	82.8	[38]
Alginate–HPMC	Avocado	Frequency: 1706 Hz	323–416	Freeze-drying	68	ND	[39]
Alginate–HMP	Roselle seed	Inner/Outer nozzle: 300/400 µm Frequency: 300 Hz Air pressure: 600 mbar	ND	Oven-drying	ND	95	[40]
Alginate–HMP	Kenaf seed	Inner/Outer nozzle: 150/300 µm Inner/Outer rate: 0.2/7 mL/min Air pressure: 600 mbar	900 (500 dried)	Air-drying–Freeze-drying	33–67	ND	[41,42]
Alginate–HMP	Kenaf seed	Inner/Outer nozzle: 200/300 µm Frequency: 500 Hz	700–920 (330–500 dried)	Freeze-drying	63	ND	[43]
Alginate- low or high amylose content starch	Canola	Inner/Outer rate: 30/200 mL/h Outer nozzle: 200 µm Frequency: 1750 Hz	310–380	pH treatment Freeze-drying	ND	58	[44]
HMP alginate/chitosan	Kenaf seed	Inner/Outer nozzle: 200/300 µm Inner/Outer rate: 0.2/7 mL/min Frequency: 500 Hz	475–775	Oven-drying (50 °C for 2 h)	33–65	ND	[45]
Alginate-HMP-chitosan	Kenaf seed	Inner/Outer nozzle: 200/300 µm Frequency: 500 Hz	ND	Freeze-drying	ND	ND	[46]
Alginate/κ-carrageenan/ chitosan	Ginger oil	Inner/Outer nozzle: 450/900 µm Frequency: 40 Hz Air pressure: 400 mbar	1600 µm	None	85	76	[47]

ND: not determined. EE: encapsulation efficiency. Load: loading. HPMC: Hydroxypropyl methylcellulose. HMP: High-methoxyl pectin.

**Table 2 foods-12-01788-t002:** Oxidation indicators comparison of free oil and encapsulated oil produced by coextrusion.

Shell and/or Coating Material	Oil/Essential Oil	Antioxidant	Oxidation Indicators		Storage Conditions	References
			Free Oil/Essential Oil	Encapsulated		
Alginate	Olive	Caffeic acid (300 ppm)	PV> 16 meq/kg p-AV > 3.2 FFA > 0.12%	PV < 14 meq/kg p-AV < 2.5 FFA <0.14%	30 days at 37 °C	[31]
Alginate	Canola	Quercetin (200 ppm)	PV> 16 meq/kg p-AV > 3.90 FFA > 0.21%	PV < 10.2 meq/kg p-AV < 2.99 FFA < 0.23%	60 days at 38°C	[32]
Alginate	Fish oil	/	PV> 11.3 meq/kg, p-AV > 11.7 DHA loss 2.72% EPA loss 0.84%	PV < 4.8 meq/kg, p-AV < 6.7 DHA reduction 0.68% EPA reduction 0.74%	17 days at 37 °C	[34]
Alginate	Linseed	/	PV > 65 meq/kg, p-AV > 9.33 FFA > 1.22	PV < 48.26 meq/kg, p-AV < 5.06 FFA < 1.26	4 weeks at 40 °C	[37]
Alginate	Rapeseed	/	PV > 65 meq/kg p-AV > 11.30 FFA > 0.30	PV < 20.77 meq/kg p-AV < 6.99 FFA 0.32	4 weeks at 40 °C	[37]
Alginate	Ginger	/	PV > 23 meq/kg p-AV > 34 TBARS: 5.8 mg MDA/kg	PV > 21 meq/kg p-AV > 40 TBARS: 5.8 mg MDA/kg	15 days at 4 °C	[47]
Alginate–HPMC	Avocado	Phloridzin or BHT (300 ppm)	Totox: 25.4	Totox: 18.0/19.7	90 days at 37 °C	[39]
Alginate–HMP	Roselle seed	/	PHY loss: 35% TOCO loss: 34.6%	PHY loss: 13.78% TOCO loss: 87.6	24 days at 65 °C	[40]
Alginate–HMP	Kenaf seed	/	PV> 11.3 meq/kg, p-AV > 11.7 FFA > 1.6	PV < 4.8 meq/kg, p-AV < 6.7 FFA < 1.41	24 days at 65 °C	[41,42]
Alginate–HMP	Kenaf seed	/	PHY loss: 59.7% TOCO loss: 51.2%	PHY loss: 32.8% TOCO loss: 12.9%	24 days at 65 °C	[43]
Alginate–low- or high-amylose-content starch	Canola	Quercetin, Vitamin E or BHT (200 ppm)	PV> 16.4 meq/kg p-AV > 3.90 FFA > 0.22%	PV < 10.8 meq/kg, p-AV < 4.19 FFA < 0.19%	60 days at 38 °C	[44]
Alginate–κ-carrageenan	Ginger	/	PV > 23 meq/kg p-AV > 34 TBARS: 5.8 mg MDA/kg	PV > 16 meq/kg p-AV > 25 TBARS: 4.86 mg MDA/kg	15 days at 4 °C	[47]
Alginate–κ-carrageenan–chitosan	Ginger	/	PV > 23 meq/kg p-AV > 34 TBARS: 5.8 mg MDA/kg	PV > 15 meq/kg p-AV > 16 TBARS: 4.44 mg MDA/kg	15 days at 4 °C	[47]
Alginate–HMP–chitosan	Kenaf seed	/	PV> 10.1 meq/kg, p-AV > 20.2 FFA > 2.26%	PV < 3.9 meq/kg, p-AV < 15.94 FFA < 1.72%	24 days at 65 °C	[46]
Alginate–chitosan	Ginger		PV > 23 meq/kg p-AV > 34	PV > 19 meq/kg p-AV > 26 TBARS: 5.2 mg MDA/kg	15 days at 4 °C	[47]

PV: peroxide value, p-AV: p-anisidine value, FFA: Free Fatty Acid, BHT: butylated hydroxytoluene, HMP: high-methoxyl pectin, HPMC: hydroxyopropyl methylcellulose, DHA: docosahexaenoic acid EPA: eicosapentaenoic acid. TOCO: tocopherols. Totox: total oxidation value. TBARS: thiobarbituric acid reactive substances.

## Data Availability

Not applicable.

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
