# Peer review of "Alginate Based Core–Shell Capsules Production through Coextrusion Methods: Recent Applications"

_foods, 2023, doi:10.3390/foods12091788_

Round 1

Reviewer 1 Report

In the present work, the authors carried out a review on the applications of encapsulation by coextrusion for the preservation of oils and essential oils, probiotics and other bioactives. The article brings a lot of relevant information and will certainly contribute to the development of this area of knowledge. However, some improvements need to be made.

1 – The introduction can be enriched with more data from the literature.

2- The sections must be properly numbered for better understanding. Some titles do not have an index.

3- The section dealing with other bioactives is poor and needs to be better described. I suggest adding other ingredients and describing them as it was done for essential oils and probiotics. Certainly, these changes will make the review more complete!

These changes must be made for the article.

Reviewer 2 Report

Alginate-based core-shell microparticles formed by co-extrusion technique are collected in this review paper. This paper is an interesting and useful collection on data of alginate-based microcapsules.

Major remarks:

1.      The title of review paper is about alginate-based core-shell capsules, however, some chitosan-based capsules and one Agar-fish gelatin-κ-carrageenan without alginate are also included. I suggest to delete them.

2.      Definition of PV and p-AV should be added to explain their significance in the characterization of oils.

3.      PV unit is meq/kg and not mq/kg

Minor remarks:

Acidophilus NCFM should be defined.

Page 2, line 47: „Such formulation, are more likely” must be corrected to „Such formulation is more likely”

Page 9, line 227: „…for cells with and excellent permeability…” should be corrected to „…for cells with excellent permeability…”

Page 10, line 255: „…obtained promising for rapid extraction…” should be corrected to „…obtained promise for rapid extraction…”

Page 10, line 257: „Capsules is suitable...” must be corrected to „Capsules are suitable…”

The meaning of bbreviations (e.g. PV, p-AV, BHT, etc.) should be included at the first presence in the paper.

Page numbering is wrong from page 8.
